# Thermal Properties of Novel Phase-Change Materials Based on Tamanu and Coconut Oil Encapsulated in Electrospun Fiber Matrices

**Evdoxia Paroutoglou** [1,*]**, Peter Fojan** [2] **, Leonid Gurevich** [2] **and Alireza Afshari** [1]

1 Department of Energy Performance, Indoor Environment and Sustainability of Buildings, Aalborg University, BUILD, 2450 København SV, Denmark; aaf@build.aau.dk

2 Department of Materials and Production, Aalborg University, 9220 Aalborg Øst, Denmark; fp@mp.aau.dk (P.F.); lg@mp.aau.dk (L.G.)

* Correspondence: evp@build.aau.dk

**Abstract:** The accumulation of thermal energy in construction elements during daytime, and its release during a colder night period is an efficient and green way to maintain a comfortable temperature range in buildings and vehicles. One approach to achieving this goal is to store thermal energy as latent heat of the phase transition using the so-called phase-change materials (PCMs). Vegetable oils came recently into focus as cheap, widely available, and environmentally friendly PCMs. In this study, we report the thermal properties of PCMs based on tamanu and coconut oils in three configurations: pure, emulsion, and encapsulated forms. We demonstrate the encapsulation of pure coconut- and tamanu-oil emulsions, and their mixtures and mixtures with commercial PCM paraffins in fiber matrices produced by a coaxial electrospinning technique. Polycaprolactone (PCL) was used as a shell, the PCM emulsion was formed by the studied oils, and sodium dodecyl sulfate (SDS) and polyvinyl alcohol (PVA) were used as emulsifiers. The addition of commercially available paraffin RT18 into a 70/30 mixture of coconut and tamanu oil, successfully encapsulated in the core of a PCL shell, demonstrated latent heats of melting and solidification of 63.8 and 57.6 kJ/kg, respectively.

**Keywords:** PCM; coconut oil; tamanu oil; electrospun fiber matrix; encapsulation; DSC

## 1. Introduction

According to the International Energy Agency [1], energy consumption in the building sector reached 36% of the total global energy use in 2018. In recent years, energy-saving technologies such as thermal energy storage (TES) systems have come into focus. TES systems are classified as sensible heat storage (SHS), latent heat storage (LHS), and thermochemical heat storage (THS) [2]. Latent heat thermal energy storage (LHTES) with phase-change materials (PCM) is currently a preferred energy storage solution due to the increasing cost of energy resources [3]. In LHTES systems, energy is absorbed as latent heat of the phase transition, and then stored and dissipated into the environment. Novel heat exchangers employing encapsulated PCMs can minimize energy use compared to traditional TES solutions [4]. The main challenge in TES technology is selecting a suitable PCM on the basis of its parameters, e.g., density, thermal conductivity, supercooling degree, phase change temperature, and heat of fusion [5]. PCM might rely on solid–solid, solid–liquid, and liquid–gas transitions. The classification of solid–liquid PCMs is presented in Figure 1.

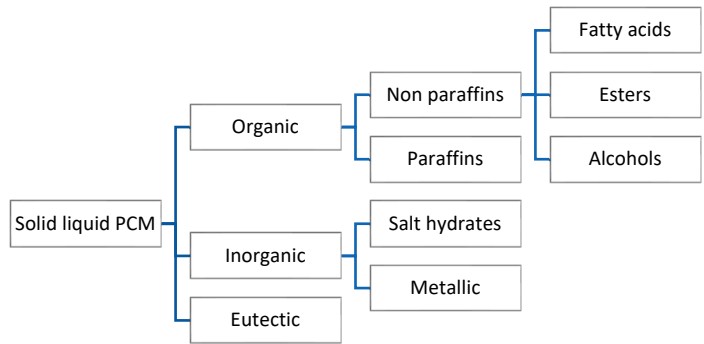

**Figure 1.** Classification of solid–liquid PCMs [6].

Organic fatty acids are abundant, sustainable, nontoxic, chemically and physically stable PCMs with relatively low cost that exhibit lower melting points than those of organic paraffins [7]. However, ingenious strategies, e.g., additives and flame retardant elements, are applied to eliminate the moderate–high flammability of organic paraffins and fatty acids [8]. Vegetable oils have recently been investigated as a replacement for traditional PCMs, such as organic paraffins and inorganic salt hydrates. Eutectic binary mixtures of fatty acids are suitable for thermal comfort applications due to possibility to adjust the melting temperature to the desired range. Arsana et al. [9] studied corn oil as a PCM applied to the walls of cooling equipment for food storage, and they achieved an energy reduction of 4% and a coefficient of performance (COP) increase of 6%. Rasta et al. [10] investigated a water-based PCM solution with the addition of soya oil ester for low-temperature TES applications. With the addition of 12.5% soya oil ester in tap water, the freezing point was decreased to $-6\,°C$, and the supercooling degree was reduced by 1 K [10]. Djuanda et al. [11] examined virgin coconut oil (VCO) with the addition of 20% soybean oil for TES applications, and a latent heat of fusion of 94.31 kJ/kg was obtained. The addition of 5% to 10% of vegetable oil esters in water with soya oil/corn oil mixtures for cold thermal energy storage (CTES) applications resulted in melting latent heat in the range of 171.72–230.68 J/g [12]. Kahwaji et al. [13] investigated the thermal properties of edible oils margarine and shortening, refined coconut oil, and virgin coconut oil as PCMs for a TES application. Coconut oil exhibited thermally stable performance with a latent heat of $105 \pm 11$ J/g and a phase-change temperature of $24.5 \pm 1.5\,°C$ [13].

Coconut oil is a renewable alternative to traditional PCMs due to its abundance, low cost, and chemical stability. According to the literature, coconut oil exhibits a liquid–solid transition at 24–27 °C [13], rendering it suitable for application in LHTES systems in a moderate climate. The thermophysical parameters of coconut oil PCMs are within the range of human thermal comfort conditions; their application reduces energy consumption and improved the temperature comfort. Faraj et al. [14] demonstrated a 53.7% increase in the span of charging and discharging processes of an underfloor electric heating system with PCM plates of coconut oil. Coconut oil is utilized in passenger vehicles [15] to reduce the cabin's average temperature by 15 °C and improve the passengers' thermal comfort. Irsyad et al. [16] studied a room cooling application with coconut oil as PCM in a container. The experimental solidification phase occurred at 24 °C, and the melting phase was reported at 22–24 °C [16]. Silalahi et al. [17,18] studied coconut oil for potential use in sensible and latent TES for room-temperature conditioning applications in Indonesia, and the latent heat of coconut oil was decreased by adding graphite nanoparticles, CuO, and ZnO. Putri et al. [19] analyzed coconut oil's thermal performance, and found that the air temperature was decreased by 2 °C for 2 kg of coconut oil in the solid–liquid phase change. In the study by Rahayu et al. [20], coconut oil with 50% lauric acid demonstrated a melting temperature of 26 °C and latent heat of 103 kJ/kg. Sutjahja et al. [21] proved that adding 1 wt.% of graphite, CuO, or ZnO to coconut oil resulted in thermal conductivity enhancement. Wonorahardjo et al. [22,23] proposed the addition of coconut oil in the building envelope to improve the thermal mass effect and air circulation. In [23], Wonorahardjo et al.

showed that 135–170 kg of coconut oil in a 3–4 m$^2$ room could decrease the air temperature by 2.0–2.5 K in the afternoon. Alqahtani et al. [24] examined the addition of a 4 cm coconut oil layer in the wall thickness as an optimal solution for improving heat storage capacity. The encapsulation of PCM in solid matrices can drastically simplify their application in construction elements. Several studies reported PCM applications of encapsulated coconut oil [25–30]. The encapsulation of coconut oil in a biochar matrix [25] for thermal insulation resulted in a maximal latent heat of 74.6 kJ/kg. A coaxial electrospinning technique was reported by Ranodhi Udangawa et al., and coconut oil was encapsulated in the core of a cellulose shell [26]. Coconut oil encapsulated by electrospinning can be successfully used for thermoregulation in the temperature range from 7 to 22 °C. Oktay et al. [27] encapsulated coconut oil with two different methods: microencapsulation and UV curing. The melting enthalpy of such microencapsulated oil was 12% higher compared to that of pure coconut oil. Spherical microcapsules with 81.1% encapsulated coconut oil was prepared by Németh et al. [28], and a stable core-shell structure was prepared. Few other studies focused on the development of microcapsules for thermoregulatory textiles [29,30]. Sarac et al. [29] developed organic fabrics with microencapsulated coconut oil in polymer shells and acquired latent heats from 6.7 to 14.9 kJ/kg.

Tamanu oil is a vegetable-based oil extracted from the seeds of *Calophyllum inophyllum* seed oil found in abundance in East Africa, southern coastal India, Malaysia, and Australia, and exhibits liquid–solid transition in the range of 13–14 °C [31]. Tamanu oil's high content in oleic and linoleic fatty acids renders it suitable for biofuel production. Several studies [32–36] addressed tamanu oil (*Calophyllum inophyllum* seed oil) potential for biodiesel production, and its composition [37], but the thermophysical properties of tamanu oil have not been characterized.

In the present paper, we report the thermal properties of coconut and tamanu oils in their pure and emulsion form, and develop a method for their encapsulation into electrospun fiber matrices. The novel fiber matrices of oils mixtures with commercial PCM paraffins produced by a coaxial electrospinning technique were characterized. The current work aims to identify the thermal properties of electrospun nanofibers as an alternative to PCMs in liquid–solid form. In future work, a layer of nanofibers containing PCM could replace traditional PCMs in LHTES applications.

## 2. Materials and Methods

### 2.1. Thermophysical Studies

The current study examined coconut oil from coconut palm tree *Cocos nucifera* and tamanu oil from *Calophyllum inophyllum* seeds as novel PCMs, and their thermal properties were analyzed. Tamanu oil and coconut oil are purchased as cosmetic and edible products. The materials' thermal properties were measured using a differential scanning calorimeter (DSC) Q200 (TA Instruments) with T-zero thermocouples. The blank reference sample was a conventional empty aluminum crucible [38]. Nitrogen and air were used as inert purge gases with a 50 mL/min controlled flow rate. The thermal cycles for tamanu oil in its first thermal cycle and coconut oil were set in the range from −30 to 80 °C with a scanning rate of 1.5 °C/min in a dynamic mode. The mass of each sample was 6 mg. The long-term performance of tamanu oil was evaluated under 50, 100, 150 and 200 thermal cycles [38] in the range from −80 to 200 °C with a scanning rate of 10 °C/min in a dynamic mode. The mass of the four samples was in the range from 5.5 to 7.5 mg.

### 2.2. PCM Emulsions and Mixtures

Two approaches were used to prepare emulsions. In the first approach, coconut oil and tamanu oil were mixed with water in a ratio of 1:1 to 5:1 in the presence of a surfactant. Table 1 shows the oil-in-water emulsions prepared using polyethylene glycol sorbitan (Tween 20) and polyoxyethylensorbitan oleate (Tween 80) as emulsifiers. The high heat capacity of water (335 kJ/kg) further contributes to the latent heat of PCM mixtures. Water

is used in phase-change materials applications for shifting the melting or solidification point [3].

**Table 1.** Mixture percentage.

| PCM | Mixing Ratio Water:(Oil:Emulsifier) | Emulsifier |
|---|---|---|
| Coconut oil | 1:(1:1)2<br>1:(1:1)5<br>1:(2:1)3 | 5% *v/v* Tween 20 |
| | 1:(4:1)3<br>1:(2:1)4<br>1:(3:1)4<br>1:(5:1)4<br>1:(1:1)5<br>1:(2:1)5 | 5% *v/v* Tween 80 |
| Tamanu oil | 1:(5:1)3<br>1:(5:1)4<br>1:(5:1)5 | 5% *v/v* Tween 20 |
| | 1:(1:1)3<br>1:(2:1)3<br>1:(3:1)3<br>1:(5:1)3<br>1:(3:1)4<br>1:(4:1)4<br>1:(2:1)5<br>1:(1:1)2 | 5% *v/v* Tween 80 |
| **PCM** | **Mixing Ratio (PCM:PCM)** | **Emulsifier** |
| Coconut oil–tamanu oil | 90% c.o.−10% t.o.<br>70% c.o.−30% t.o.<br>50% c.o.−50% t.o.<br>30% c.o.−70% t.o.<br>10% c.o.−90% t.o. | - |

In the second approach, the two oils were mixed in several ratios (Table 1) without emulsifiers, and the thermal properties of these new solutions were evaluated. The thermal behavior of all mixtures was investigated by DSC using DSC Q2000.

Tamanu and coconut oil/water emulsions in ratios 1:1 to 5:1 were prepared with polyethylene glycol sorbitan (Tween 20) and polyoxyethylensorbitan oleate (Tween 80) as emulsifiers. The structure of emulsions was imaged with an Axioscop plus 2 optical microscope, and analyzed with ImageJ [39]. The mixtures that had not been phase-separated were analyzed using the DSC technique. The thermal cycles for all emulsions were set in the range from −30 to 80 °C with a scanning rate of 1.5 °C/min in a dynamic mode. The mass of each sample was 6 mg. The final expanded uncertainty is 0.1% for each experimental measurement.

*2.3. PCM Electrospun Fiber Matrix*

Electrospun fibers were produced by coaxial electrospinning using a 2.2.D−500—Yflow electrospinner. The inner diameters of the outer and inner needles were 20 and 10 G, respectively. The voltage threshold for the electrospinning process was in the range from 10 to 15 kV. The process yielded fibers consisting of a PCM in the core encapsulated into solid polymeric shell.

The shell for all the produced fibers was composed of either 9% *w/v* or 12% *w/v* PCL dissolved in dichloromethane (DCM). The combination of studied concentrations and flow rates is shown in Table 2.

**Table 2.** Cases examined for the construction of PCM electrospun fiber matrices.

| Case | Sheath Solution Concentration (%*w/v*) | Flow Rate (mL/h) Core Solution | Flow Rate (mL/h) Sheath Solution |
|---|---|---|---|
| 1st | 9% | 0.3 | 0.6 |
| 2nd | 9% | 0.5 | 0.5 |
| 3rd | 12% | 0.5 | 0.5 |

Three different types of core materials were studied: the water emulsions of pure tamanu and coconut oil, a mixture of tamanu and coconut oils, and different mixtures of tamanu and coconut oils with commercial PCM materials, as shown in Table 3.

**Table 3.** PCM compositions encapsulated in the fiber core.

| PCM | Tamanu Oil (t.o.) | Coconut Oil (c.o.) | Tamanu Oil (t.o.) and Coconut Oil (c.o.) | | Tamanu Oil (t.o.) and Coconut Oil (c.o.) and Commercial PCM | | |
|---|---|---|---|---|---|---|---|
| % | 100% (t.o.) | 100% (c.o.) | 30% (t.o.) | 70% (c.o.) | 15% (t.o.) | 35% (c.o.) | 50% RT15 |
| | | | | | 15% (t.o.) | 35% (c.o.) | 50% RT18 |
| | | | | | 15% (t.o.) | 35% (c.o.) | 50% PT15 |
| | | | | | 15% (t.o.) | 35% (c.o.) | 50% PT18 |

Oil/water emulsion samples were prepared by mixing a 10% *w/v* PVA solution with an 80% *v/v* PCM/water emulsion (with 8.4 mmol/L sodium dodecyl sulfate added as a surfactant) in 1:1 ratio. The resulting PCM–PVA emulsion was homogenized using a T25 digital ULTRA-TURRAX disperser with 0.07% of a nonionic surfactant Triton X added to improve emulsion's spinnability.

For the third type of samples, four commercial PCMs (RT15, RT18, PT15, and PT18) were mixed with tamanu and coconut oils in the ratios shown in Table 3, and were encapsulated in a PCL shell by electrospinning. The commercially available PCMs (RT15 and RT18 from Rubitherm Technologies GmbH and PT15 and PT18 from Pure Temp LLC) were analyzed earlier in [38,40].

The obtained electrospun fiber matrices were characterized with DSC and scanning electron microscopy (SEM) using a Zeiss XB1540 field-emission electron microscope. The thermal cycles for all fiber matrices were set in the range from −30 to 80 °C with a scanning rate of 1.5 °C/min in a dynamic mode. The mass of each sample was 3 mg.

## 3. Results and Discussion

### 3.1. Thermophysical Characterization of Pure PCMs

The DSC thermograms of tamanu oil for 200 thermal cycles and coconut oil for the first thermal cycle are shown in Figure 2. The obtained melting or solidification temperatures and enthalpies are displayed on the graphs; for each experimental measurement, the expanded uncertainty was ±0.1%.

The thermal properties and stability of tamanu oil were characterized with DSC by thermal cycling. As can be seen in Figure 2a, thermograms for the 1st, 50th, 100th, 150th, and 200th thermal cycles perfectly coincided, and no supercooling was observed, indicating no degradation or other changes in material properties. Tamanu oil exhibited average melting and solidification peaks at 0.86 °C, and the average latent heat of melting and solidification were 3.56 and 4.64 kJ/kg.

Several studies [15,18,41,42] identified coconut oil as a suitable PCM candidate. In this work, the thermophysical properties of coconut oil were studied, and Figure 2b depicts the DSC thermograph of coconut oil in its first thermal cycle. According to the

literature [15,18,41,42], the average melting temperature and latent heat of fusion for coconut oil were in the range from 22 to 28 °C and 70 to 255 kJ/kg, respectively. The average melting and solidification peaks were at 23.05 and 7.47 °C, respectively. A thermal hysteresis was observed in coconut oil with a supercooling of 15.58 °C. The average latent heats of melting and solidification of coconut oil are 50.22 kJ/kg and 56.71 kJ/kg.

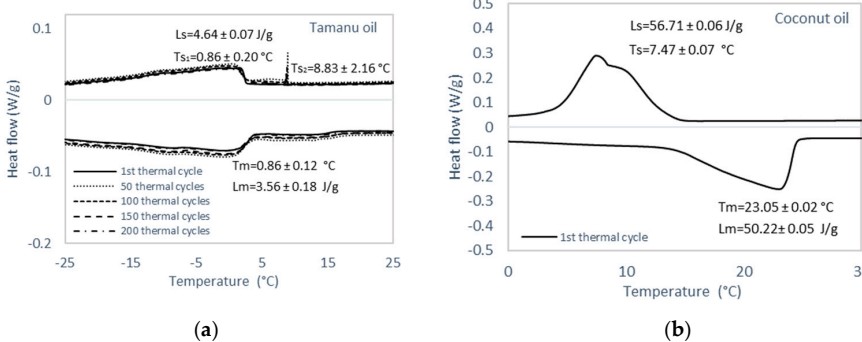

**Figure 2.** DSC thermogram of (**a**) tamanu oil in thermal cycles 1, 50, 100, 150, 200; (**b**) coconut oil in the first thermal cycle.

### 3.2. Thermophysical Characterization of PCM Emulsions

The DSC thermographs of the two oil emulsions are shown in Figure 3. Figure 3a,b show the thermograph for coconut oil/water emulsions mixed in ratios from 1:1 to 5:1 and with Tween 20 and Tween 80 as surfactants, respectively. Thermographs for similar emulsions produced with tamanu oil are shown in Figure 3c,d. Optical microscopy images of these oil/water emulsions are presented in Figures 4–7. Table 4 summarizes the DSC results and the average emulsion size obtained from optical images.

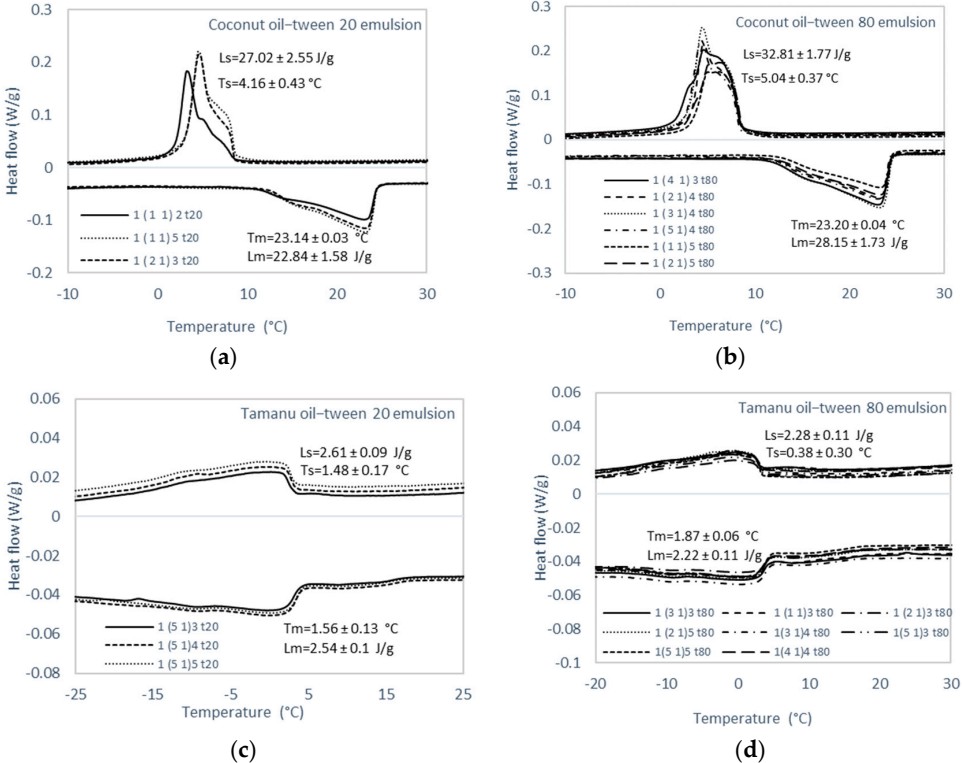

**Figure 3.** DSC thermograms of (**a**) coconut oil/water emulsions with Tween 20, (**b**) coconut oil/water emulsions with Tween 80, (**c**) tamanu oil/water emulsions with Tween 20, (**d**) tamanu oil/water emulsions Tween 80 in their first thermal cycle.

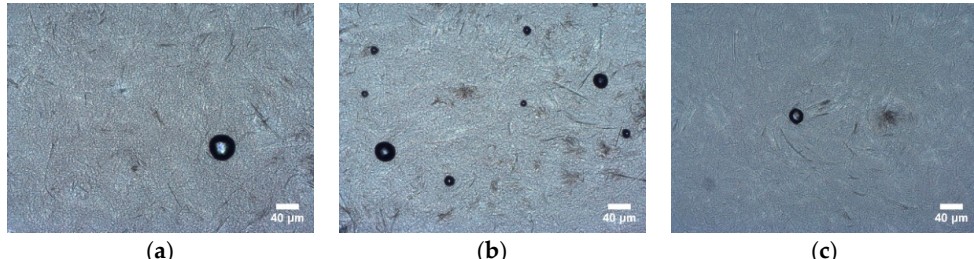

**Figure 4.** Optical microscopy images of coconut oil/(water emulsion stabilized with Tween 20), water:(oil:emulsifier): (**a**) 1:(1:1)2, (**b**) 1:(1:1)5, (**c**) 1:(2:1)3.

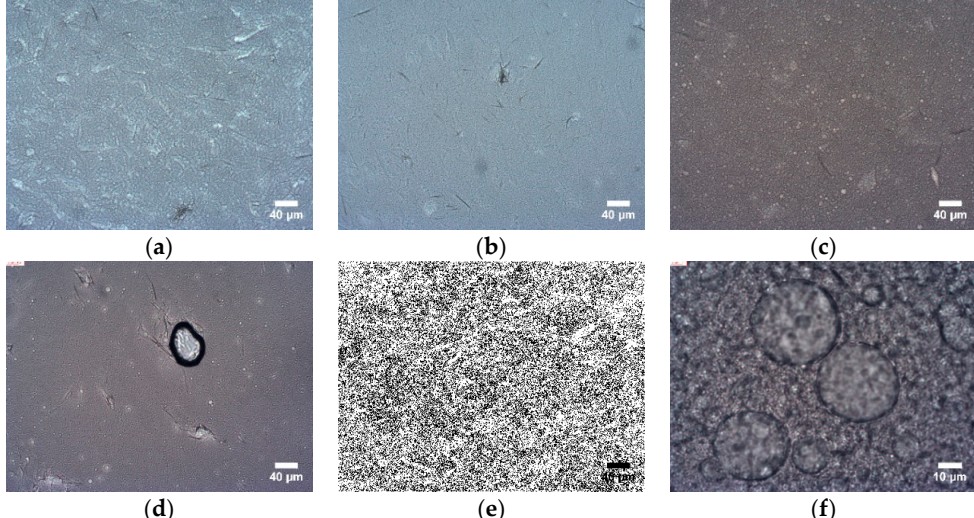

**Figure 5.** Optical microscopy images of coconut oil/(water emulsion stabilized with Tween 80), water:(oil:emulsifier); (**a**) 1:(4:1)3, (**b**) 1:(2:1)4, (**c**) 1:(3:1)4, (**d**) 1:(5:1)4,(**e**) 1:(1:1)5, (**f**) 1:(2:1)5.

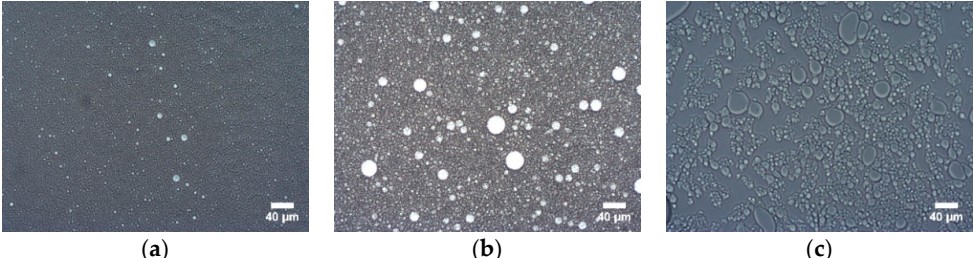

**Figure 6.** Optical microscopy images of tamanu oil/(water emulsion stabilized with Tween 20), water:(oil:emulsifier); (**a**) 1:(5:1)3, (**b**) 1:(5:1)4, (**c**) 1:(5:1)5.

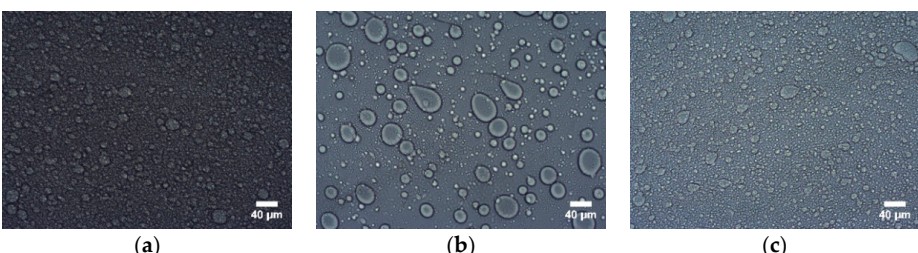

**Figure 7.** *Cont.*

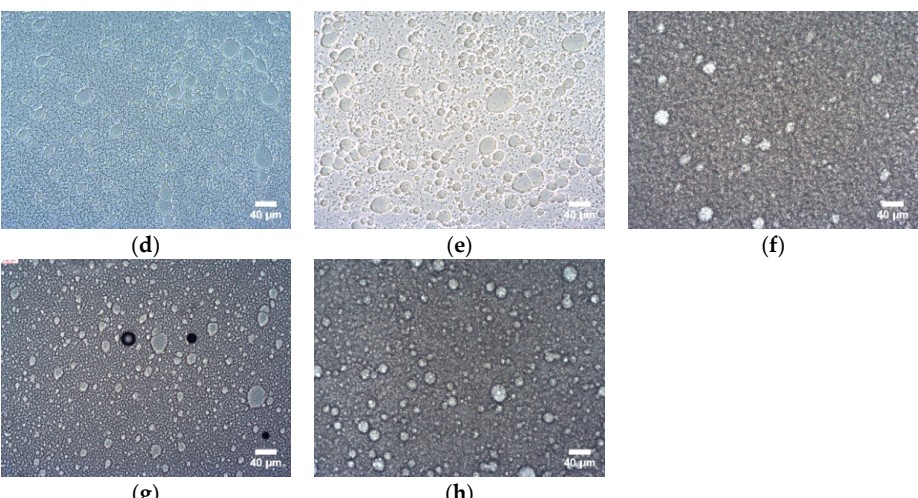

**Figure 7.** Optical microscopy images of tamanu oil/water emulsion stabilized with Tween 80, water:(oil:emulsifier); (**a**) 1:(1:1)3, (**b**) 1:(2:1)3, (**c**) 1:(3:1)3, (**d**) 1:(5:1)3, (**e**) 1:(3:1)4, (**f**) 1:(4:1)4, (**g**) 1:(2:1)5, (**h**) 1:(1:1)2.

**Table 4.** DSC analyses of coconut oil Tween 20, coconut oil Tween 80, tamanu oil Tween 20, tamanu oil Tween 80 in water emulsions in 1st thermal cycle.

| Materials | Mixing Ratio Water:(Oil: Emulsifier) | Melting Temperature (°C) | Melting Enthalpy (kJ/kg) | Solidification Temperature (°C) | Solidification Enthalpy (kJ/kg) | Average Emulsion Size (um) |
|---|---|---|---|---|---|---|
| Coconut oil Tween 20 | 1:(1:1)2 | 23.08 | 19.69 | 3.30 | 22.28 | 10.5 ± 0.9 |
| | 1:(1:1)5 | 23.19 | 24.52 | 4.56 | 31.04 | 10.1 ± 0.8 |
| | 1:(2:1)3 | 23.16 | 24.32 | 4.63 | 27.74 | 10 ± 0.9 |
| Coconut oil Tween 80 | 1:(4:1)3 | 23.08 | 31.92 | 4.58 | 37.91 | 17.4 ± 2.8 |
| | 1:(2:1)4 | 23.31 | 27.11 | 4.54 | 31.54 | 7.6 ± 0.3 |
| | 1:(3:1)4 | 23.26 | 33.29 | 4.43 | 37.27 | 12 ± 0.9 |
| | 1:(5:1)4 | 23.04 | 29.49 | 4.30 | 32.65 | 8.8 ± 0.5 |
| | 1:(1:1)5 | 23.26 | 22.13 | 6.16 | 26.15 | 4.1 ± 0.1 |
| | 1:(2:1)5 | 23.22 | 24.95 | 6.24 | 31.31 | 4.8 ± 1.2 |
| Tamanu oil Tween 20 | 1:(5:1)3 | 1.31 | 2.34 | 1.15 | 2.46 | 16.2 ± 0.7 |
| | 1:(5:1)4 | 1.63 | 2.64 | 1.69 | 2.58 | 5.6 ± 0.2 |
| | 1:(5:1)5 | 1.73 | 2.64 | 1.61 | 2.78 | 16.2 ± 0.2 |
| Tamanu oil Tween 80 | 1:(1:1)3 | 1.95 | 1.82 | 0.13 | 2.14 | 13.3 ± 0.4 |
| | 1:(2:1)3 | 1.85 | 1.82 | −0.26 | 1.97 | 8.4 ± 0.7 |
| | 1:(3:1)3 | 1.76 | 2.13 | 0.39 | 2.02 | 12.1 ± 0.3 |
| | 1:(5:1)3 | 1.97 | 2.39 | −0.10 | 2.59 | 16.6 ± 0.7 |
| | 1:(3:1)4 | 2.20 | 2.28 | 0.25 | 2.35 | 15.2 ± 0.8 |
| | 1:(4:1)4 | 1.72 | 2.16 | −0.53 | 2.60 | 11.5 ± 0.4 |
| | 1:(2:1)5 | 1.69 | 2.45 | 1.10 | 1.94 | 12.1 ± 0.3 |
| | 1:(1:1)2 | 1.81 | 2.73 | 2.06 | 2.66 | 9.1 ± 0.3 |

The DSC thermographs, optical microscopy images, and analytical results of tamanu and coconut oil mixtures in different ratios are shown in Figure 8 and Table 5, respectively. The expanded uncertainty for each experimental measurement was ±0.1%.

Maruyama et al. [42] studied the effect of commercial emulsifiers EM1 and EM2 on coconut oil, and observed an increase in melting and solidification enthalpies to 105.5 and 26.7 kJ/g upon the addition of 1% *w/w* of EM1. The peak melting and solidification temperatures [42] were 21.3 and 3.3 °C. The average melting and solidification temperatures, and solidification enthalpy of coconut oil in water emulsions with Tween 20 and Tween 80 (Figure 3a,b) values found in this study coincided with those presented in [42]. However, the melting enthalpies ranged from 19 to 33 kJ/kg (Table 4). In all coconut oil

emulsions thermograms (Figure 3a,b), a thermal shift of 17–20 °C between the melting and solidification curves was observed. Wiyani et al. [43] successfully formulated virgin coconut oil emulsions in a ratio of 80:20 with Tween 80 and Span 80 as emulsifiers. Gulao et al. [44] studied the physicochemical properties of coconut oil in water emulsions with two biopolymers, which resulted in average droplet diameters in the range from 310 to 1200 nm. The average emulsion size, analyzed in Table 4, showed that the coconut oil–Tween 20 (Figure 4) and Tween 80 (Figure 5) emulsion size varied between 9 and 10 µm, and 4 and 17 µm, respectively.

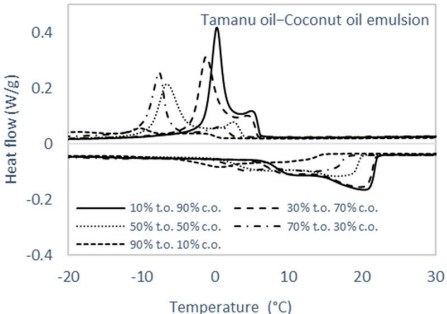

**Figure 8.** DSC thermograph of different tamanu–coconut oil mixtures in their first thermal cycle.

**Table 5.** DSC analyses of coconut–tamanu oil mixtures in their first thermal cycle.

| Materials | Mixing Ratio | Melting Temperature (°C) | | Melting Enthalpy (kJ/kg) | Freezing Temperature (°C) | | Solidification Enthalpy (kJ/kg) |
|---|---|---|---|---|---|---|---|
| | | Max | Min | | Max | Min | |
| Coconut oil–tamanu oil mixture | 90% c.o.–10% t.o. | 20.28 | 9.80 | 45.34 | 0.23 | 5.25 | 44.89 |
| | 70% c.o.–30% t.o. | 19.87 | 9.22 | 43.60 | 1.26 | 4.75 | 47.86 |
| | 50% c.o.–50% t.o. | 16.60 | 5.27 | 38.81 | −6.66 | 2.45 | 38.39 |
| | 30% c.o.–70% t.o. | 13.43 | 3.82 | 28.90 | −7.68 | 0.71 | 28.91 |
| | 10% c.o.–90% t.o. | 0.75 | 12.06 | 24.03 | −10.24 | −1.33 | 5.85 |

In [14], Isryad et al. tested *Calophyllum inophyllum* seed oil (CISO) with water, resulting in peak solidification and melting temperatures of −11.66 and 15.23 °C and latent heat of solidification and melting of 188.31 and 219.74 kJ/kg. In the current study, the melting and solidification temperatures and enthalpies of tamanu oil emulsions with Tween 20 and 80 were in the range from −0.5 to 2.3 °C, and from 2 to 3 kJ/kg. For tamanu oil emulsions (Figure 3c,d), a slight thermal shift in the range of 0.06–2.2 °C between the melting and solidification curves occurred. Urbánková et al. [45] studied tamanu and black cumin oil emulsions with the addition of sodium caseinate, and observed an average emulsion size in the range from 0.3 to 1.5 µm. In the current study, the average emulsion size of tamanu oil with Tween 20 (Figure 6) and Tween 80 (Figure 7) was 5–16 and 8–17 µm, respectively. The examined macroemulsion systems were composed of coconut oil and tamanu oil in water, and formed droplets in the micrometer range. This indicates a microemulsion that is generally thermodynamically unstable. One emulsifier was used to achieve the stability of oil in water emulsion in all different ratios, and the mixing was achieved with the same energy input.

Moreover, tamanu oil was mixed with coconut oil as pure substances without emulsifiers. In the literature, *Jatropha curcas* seed oil (JCSO) mixed with crude palm oil (CPO) [14] resulted in melting and solidification peak temperatures of −1.79 and −14.98 °C, and melting–solidification enthalpies of 8.44 and 21.17 kJ/kg. The DSC thermographs of

tamanu oil and coconut oil mixtures (Figure 8) indicate that, for all examined mixing ratios (Table 5), two peaks for tamanu oil and coconut oil could be identified in the melting and solidification thermal cycles. A hysteresis loop of the latent heat/temperature thermographs is illustrated in tamanu oil mixtures with coconut oil. The higher the percentage of coconut oil in the solution was, the higher the latent heat of fusion in the mixture. More precisely, in the mixture containing 90% coconut oil and 10% tamanu oil, there was an increase of 89% in the latent heat of melting and 667% in the latent heat of solidification compared to the mixture with 10% coconut oil and 90% tamanu oil.

### 3.3. Thermophysical Characterization of PCM Electrospun Fiber Matrix

Tamanu oil, coconut oil, and their mixtures with commercially available organic PCM (Table 3) were encapsulated in the core of electrospun fiber matrices. The obtained fiber mats were cut to prepare samples of 3 mg each. Figure 9 and Table 6 display the thermographs acquired by DSC and the summary of results, respectively. Figures 10–12 show the optical images of the PCM fiber matrices and the respective histograms of fiber diameters. The encapsulation ratio, efficiency, and mean diameter of the electrospun PCM fiber matrices are summarized in Table 7. Equations (1) and (2) were used for the calculation of the encapsulation ratio and efficiency presented in Table 7.

$$n = \frac{L_{m,encap.PCM}}{L_{m,PCM}} \tag{1}$$

$$\varepsilon = \frac{L_{m,encap.PCM} + L_{s,encap.PCM}}{L_{m,PCM} + L_{s,PCM}} \tag{2}$$

**Table 6.** DSC analyses of electrospun PCM fiber samples in 1st thermal cycle.

| Fibers | PCL | Flow Rate | Melting Temperature (°C) | | Enthalpy (J/g) | | Freezing Temperature (°C) | | Enthalpy (J/g) | |
|---|---|---|---|---|---|---|---|---|---|---|
| | | | PCM | PCL | PCM | PCL | PCM | PCL | PCM | PCL |
| Coconut oil | 9% | 0.3–0.6 mL/h | 56.55 | - | 34.42 | - | 34.47 | - | 39.08 | - |
| | | 0.5–0.5 mL/h | 22.94 | 54.71 | 46.91 | 7.36 | 8.03 | 36.49 | 40.16 | 8.59 |
| | 12% | 0.5–0.5 mL/h | 22.99 | 54.80 | 49.20 | 6.87 | 7.41 | 36.48 | 46.50 | 8.10 |
| Tamanu oil | 9% | 0.3–0.6 mL/h | −0.16 | 53.60 | 2.59 | 14.39 | 0.11 | 33.76 | 3.41 | 14.62 |
| | | 0.5–0.5 mL/h | −0.12 | 50.96 | 2.35 | 5.98 | −0.05 | 30.21 | 3.01 | 6.64 |
| | 12% | 0.5–0.5 mL/h | 0.26 | 50.51 | 2.83 | 11.03 | 0.12 | 31.26 | 3.32 | 10.74 |
| Coconut oil 70%–tamanu oil 30% | 9% | 0.3–0.6 mL/h | 20.73 | 55.43 | 20.03 | 19.34 | 7.75, - | 36.52 | 7.97 | 22.30 |
| | | 0.5–0.5 mL/h | 20.13 | 54.26 | 43.93 | 5.74 | 3.01, −1.26 | 34.34 | 35.32 | 7.07 |
| | 12% | 0.5–0.5 mL/h | 20.32 | 53.27 | 31.83 | 8.94 | 4.43, - | 33.33 | 27.04 | 10.19 |
| Coconut oil 35%–tamanu oil 15%–RT18 50% | 9% | 0.3–0.6 mL/h | 13.65 | 53.75 | 56.69 | 5.90 | 8.98, 4.48 | 34.91 | 37.00 | 6.86 |
| | | 0.5–0.5 mL/h | 13.65 | 53.78 | 63.80 | 7.44 | 9.02, 4.55, 1.48 | 34.85 | 57.58 | 8.07 |
| | 12% | 0.5–0.5 mL/h | 13.61 | 54.20 | 54.78 | 8.75 | 1.68, 4.60, 9.27 | 35.10 | 46.11 | 10.22 |
| Coconut oil 35%–tamanu oil 15%–RT15 50% | 9% | 0.3–0.6 mL/h | 12.96 | 54.12 | 50.79 | 9.81 | 9.12, 1.67, 13.15 | 35.03 | 30.99 | 11.46 |
| | | 0.5–0.5 mL/h | 12.99 | 53.75 | 53.01 | 7.11 | 9.46, 1.63, 12.84 | 34.92 | 31.89 | 7.61 |
| | 12% | 0.5–0.5 mL/h | 13.48 | 53.78 | 48.55 | 9.75 | 12.74, 1.52 | 34.92 | 31.85 | 10.27 |
| Coconut oil 35%–tamanu oil 15%–PT18 50% | 9% | 0.3–0.6 mL/h | 6.26 | 50.55 | 22.45 | 7.41 | −6.37 | 33.04 | 30.51 | 8.66 |
| | | 0.5–0.5 mL/h | 5.89 | 50.71 | 23.65 | 7.75 | −6.44 | 33.71 | 33.44 | 8.42 |
| | 12% | 0.5–0.5 mL/h | 5.83 | 2.69 | 17.60 | 8.65 | −6.70 | 33.59 | 28.59 | 9.52 |
| Coconut oil 35%–tamanu oil 15%–PT 15 50% | 9% | 0.3–0.6 mL/h | 8.71 | 53.99 | 30.03 | 5.58 | −0.83 | 35.55 | 46.81 | 6.99 |
| | | 0.5–0.5 mL/h | 9.06 | 54.05 | 33.80 | 4.89 | −0.81 | 34.70 | 44.64 | 5.97 |
| | 12% | 0.5–0.5 mL/h | 10.71 | 55.59 | 43 | 10.43 | 3.05, 0.16 | 35.96 | 57.96 | 13.46 |

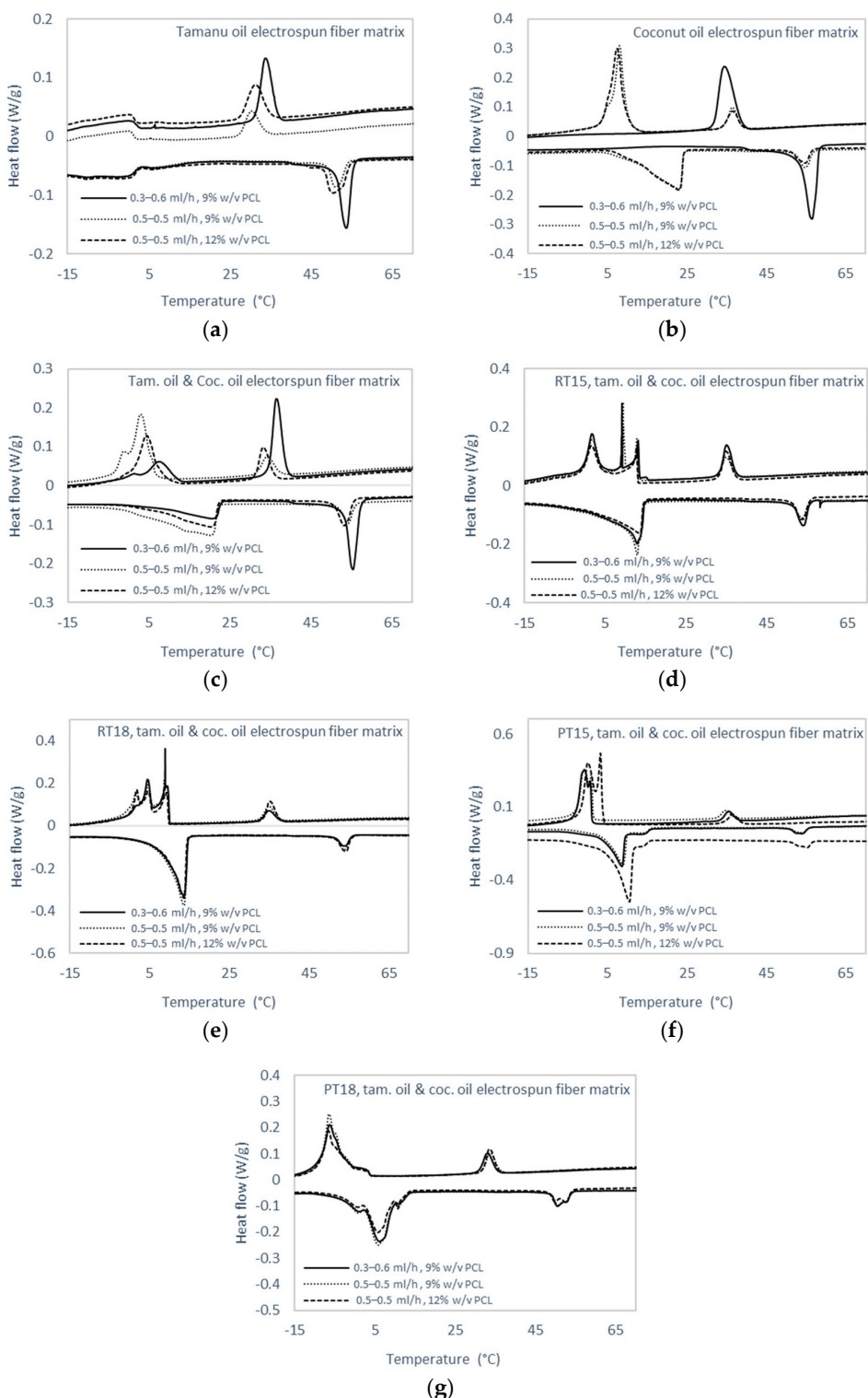

**Figure 9.** DSC thermograms of PCM encapsulated in the core of electrospun fiber in 1st thermal cycle. Core compositions: (**a**) tamanu oil emulsion; (**b**) coconut oil emulsion; (**c**) mixture of 70% co-conut oil and 30% tamanu oil; (**d**) mixture of 50% RT15, 15% tamanu oil, and 30% coconut oil; (**e**) mixture of 50% RT18, 15% tamanu oil, and 35% coconut oil; (**f**) mixture of 50% PT15, 15% tamanu oil and 35% coconut oil; (**g**) mixture of 50% PT18, 15% tamanu oil, and 35% coconut oil.

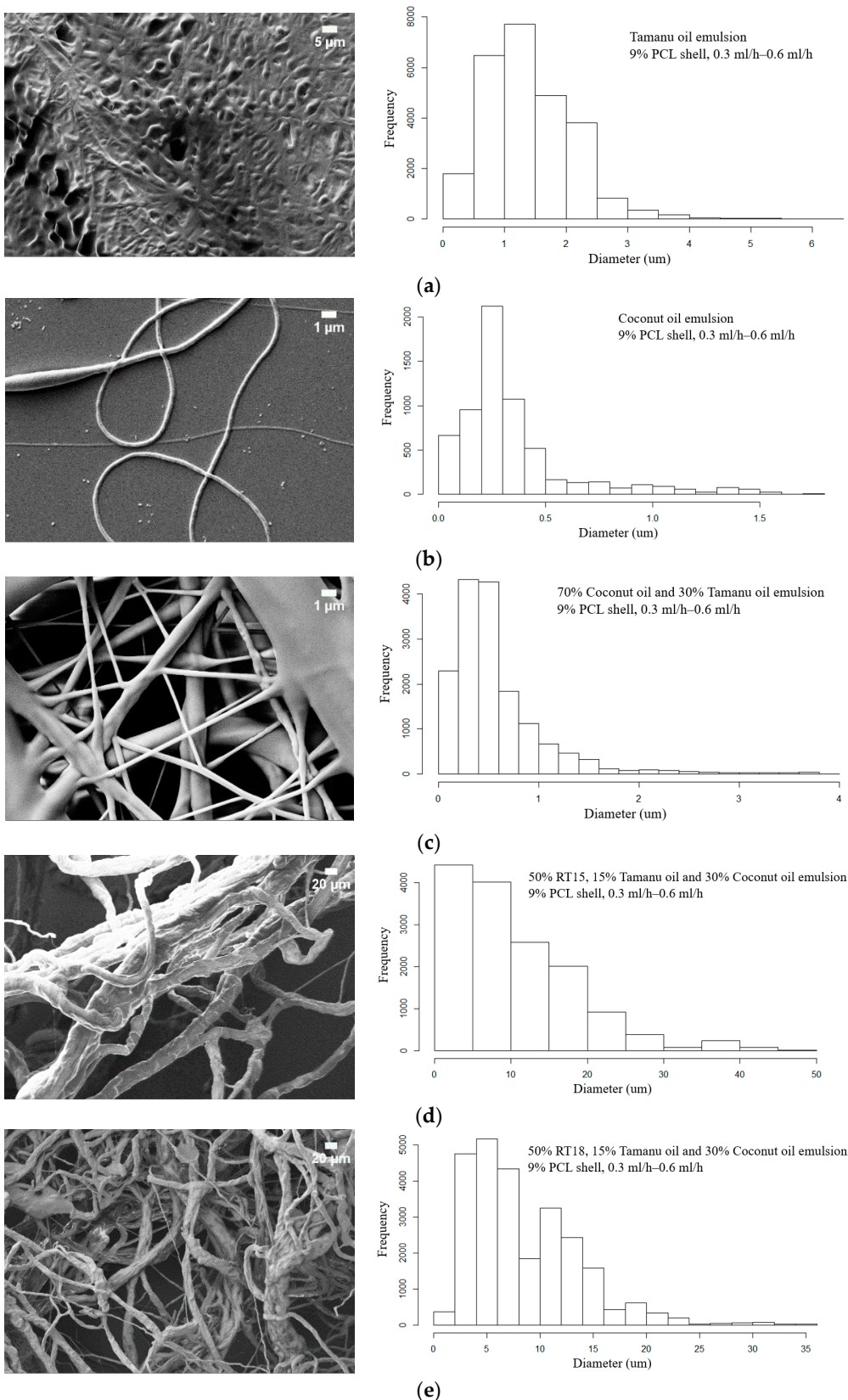

**Figure 10.** *Cont.*

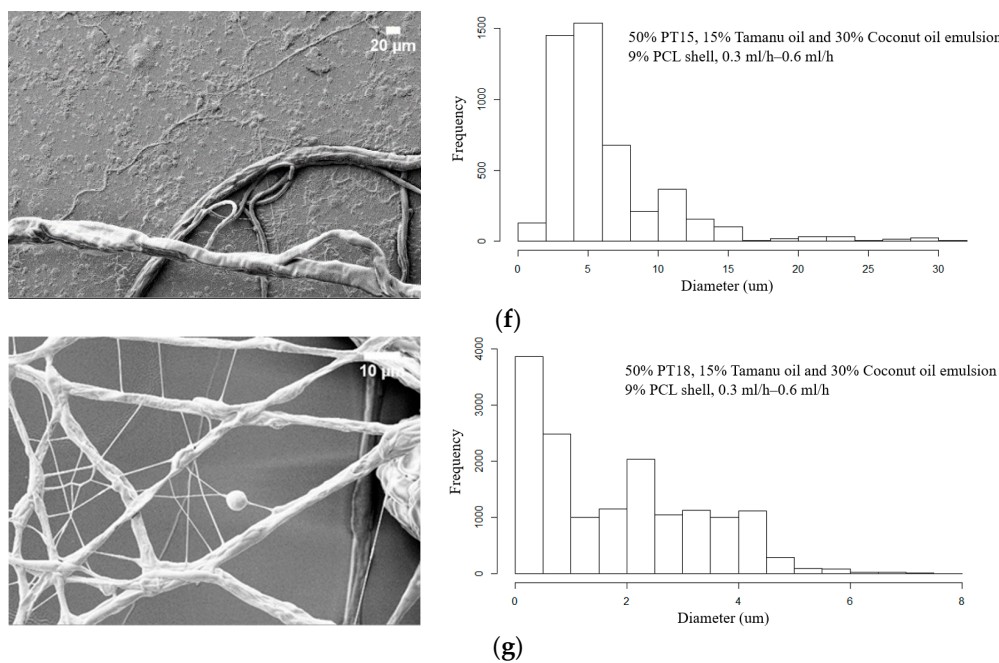

(f)

(g)

**Figure 10.** (**left**) SEM images of core/shell fibers and (**right**) the respective fiber diameter distributions for electrospun fiber mats produced with 9% PCL shell at flow rates of 0.3 mL/h for the core and 0.6 mL/h for the shell. Core compositions: (**a**) tamanu oil emulsion; (**b**) coconut oil emulsion; (**c**) mixture of 70% coconut oil and 30% tamanu oil; (**d**) mixture of 50% RT15, 15% tamanu oil, and 30% coconut oil; (**e**) mixture of 50% RT18, 15% tamanu oil, and 35% coconut oil; (**f**) mixture of 50% PT15, 15% tamanu oil and 35% coconut oil; (**g**) mixture of 50% PT18, 15% tamanu oil, and 35% coconut oil.

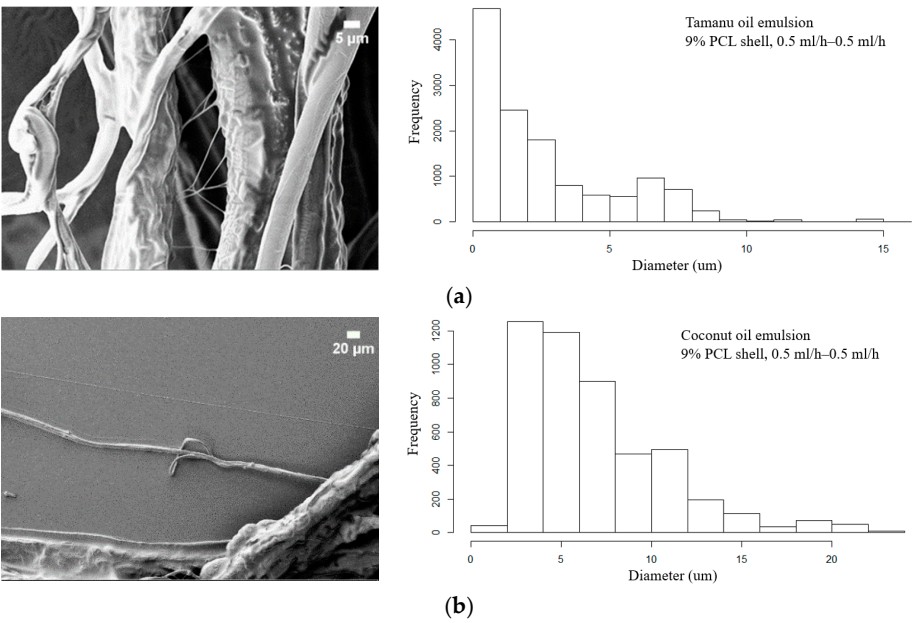

(a)

(b)

**Figure 11.** *Cont.*

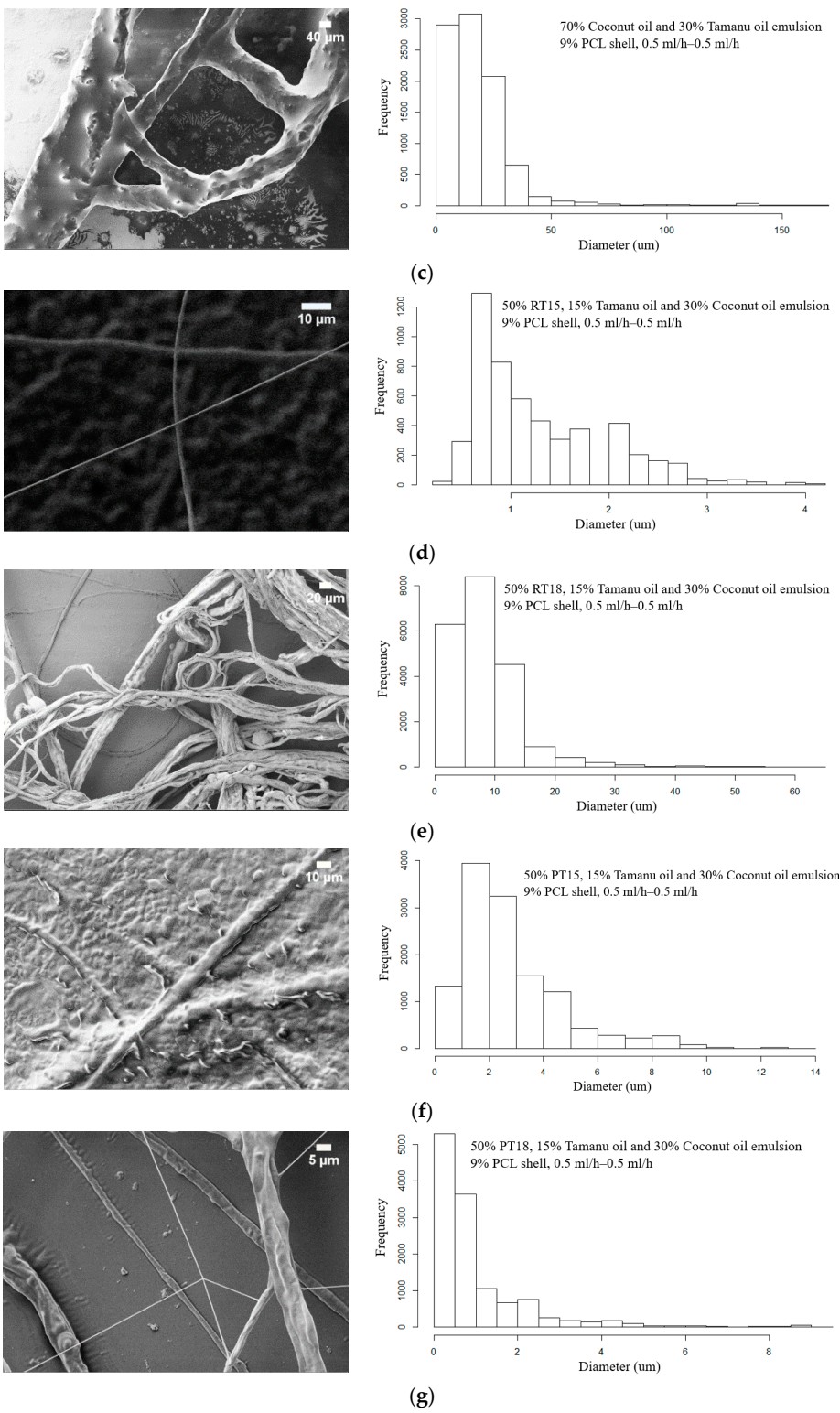

**Figure 11.** (**left**) SEM images of core/shell fibers and the (**right**) respective fiber diameter distributions for electrospun fiber mats produced with 9% PCL shell at flow rates of 0.5 mL/h for the core and 0.5 mL/h for the shell. Core compositions: (**a**) tamanu oil emulsion; (**b**) coconut oil emulsion; (**c**) mixture of 70% coconut oil and 30% tamanu oil; (**d**) mixture of 50% RT15, 15% tamanu oil, and 30% coconut oil; (**e**) mixture of 50% RT18, 15% tamanu oil, and 35% coconut oil; (**f**) mixture of 50% PT15, 15% tamanu oil, and 35% coconut oil; (**g**) mixture of 50% PT18, 15% tamanu oil, and 35% coconut oil.

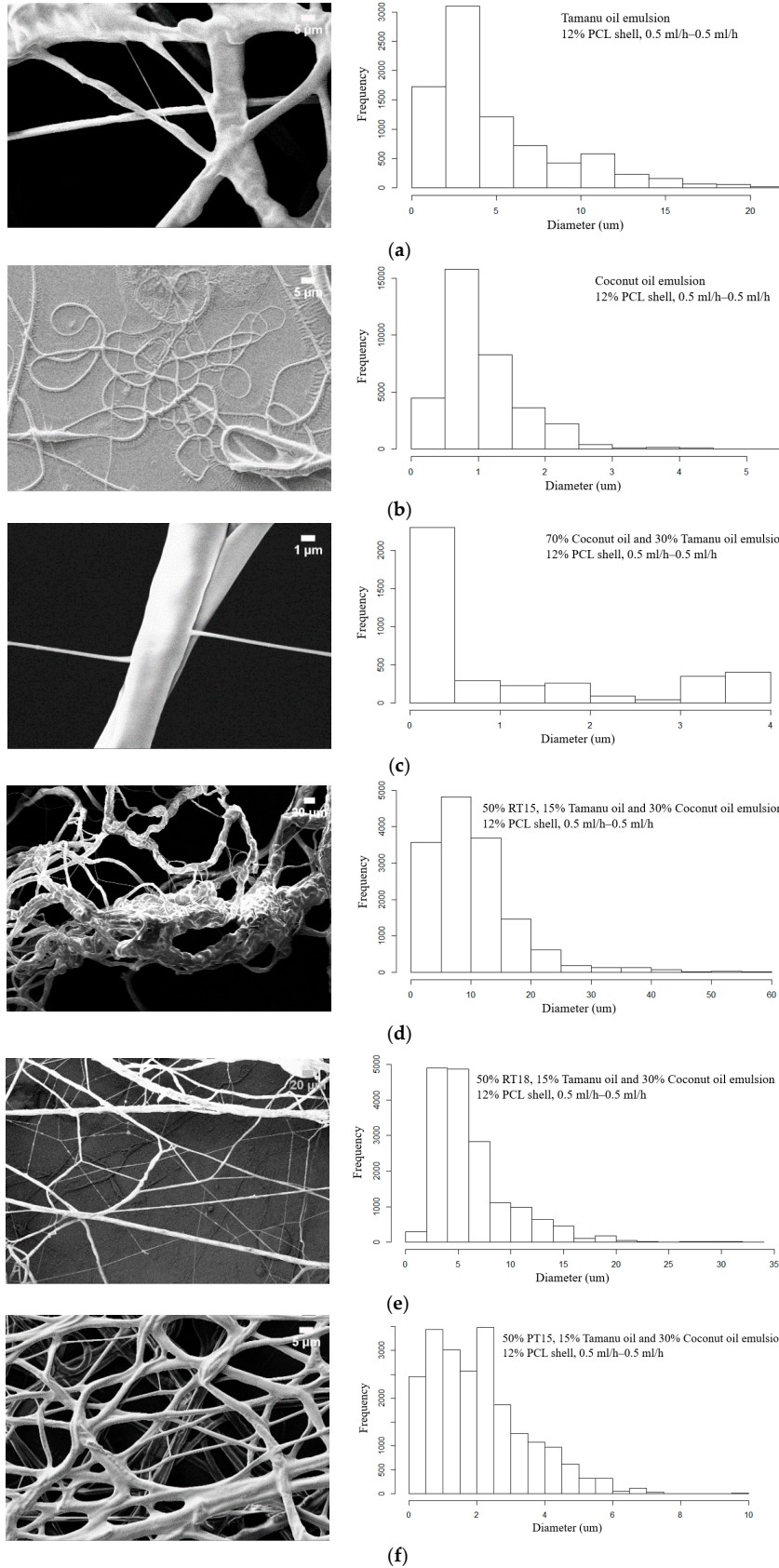

**Figure 12.** *Cont*.

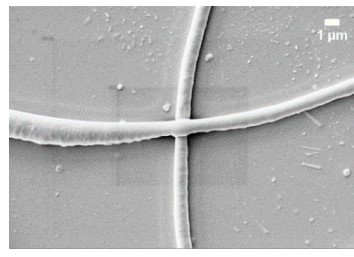 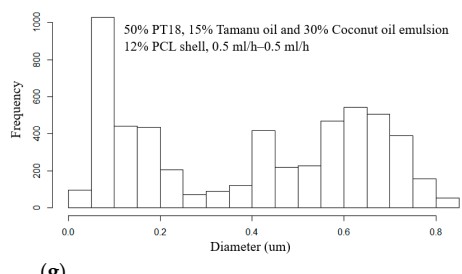

(**g**)

**Figure 12.** (**left**) SEM images of core/shell fibers and (**right**) the respective fiber diameter distributions for electrospun fiber mats produced with 12% PCL shell at flow rates of 0.5 mL/h for the core and 0.5 mL/h for the shell. Core compositions: (**a**) tamanu oil emulsion; (**b**) coconut oil emulsion; (**c**) mixture of 70% coconut oil and 30% tamanu oil; (**d**) mixture of 50% RT15, 15% tamanu oil, and 30% coconut oil; (**e**) mixture of 50% RT18, 15% tamanu oil, and 35% coconut oil; (**f**) mixture of 50% PT15, 15% tamanu oil, and 35% coconut oil; (**g**) mixture of 50% PT18, 15% tamanu oil, and 35% coconut oil.

**Table 7.** Encapsulation ratio, efficiency and mean diameter of encapsulated electrospun PCM.

| PCM Core Material | PCL | Flow Rate (mL/h) | Encapsulation Ratio n (%) | Encapsulation Efficiency ε (%) | Histogram Mean Diameter (um) |
|---|---|---|---|---|---|
| Coconut oil | 9% | 0.3–0.6 | 68.5 | 68.7 | 0.3 ± 0.001 |
| | | 0.5–0.5 | 93.4 | 81.4 | 5.6 ± 0.05 |
| | 12% | 0.5–0.5 | 98 | 89.5 | 0.9 ± 0.002 |
| Tamanu oil | 9% | 0.3–0.6 | 72.8 | 73.2 | 1.3 ± 0.06 |
| | | 0.5–0.5 | 66.1 | 65.4 | 0.8 ± 0.004 |
| | 12% | 0.5–0.5 | 79.6 | 75.1 | 2.7 ± 0.02 |
| Coconut oil 70%–tamanu oil 30% | 9% | 0.3–0.6 | 45.9 | 30.6 | 0.4 ± 0.002 |
| | | 0.5–0.5 | 100 | 86.6 | 12.8 ± 0.09 |
| | 12% | 0.5–0.5 | 73 | 64.4 | 0.2 ± 0.0004 |
| Coconut oil 35%–tamanu oil 15%–PT15 50% | 9% | 0.3–0.6 | 39.3 | 51.2 | 4.4 ± 0.03 |
| | | 0.5–0.5 | 44.2 | 52.2 | 2 ± 0.008 |
| | 12% | 0.5–0.5 | 56.3 | 67.2 | 1.5 ± 0.01 |
| Coconut oil 35%–tamanu oil 15%–PT18 50% | 9% | 0.3–0.6 | 28.1 | 34.4 | 0.5 ± 0.001 |
| | | 0.5–0.5 | 29.6 | 37.1 | 0.5 ± 0.001 |
| | 12% | 0.5–0.5 | 22 | 30 | 0.1 ± 0.0004 |
| Coconut oil 35%–tamanu oil 15%–RT15 50% | 9% | 0.3–0.6 | 100 | 96 | 6.5 ± 0.06 |
| | | 0.5–0.5 | 100 | 99.6 | 0.8 ± 0.005 |
| | 12% | 0.5–0.5 | 100 | 94.3 | 7.5 ± 0.04 |
| Coconut oil 35%–tamanu oil 15%–RT18 50% | 9% | 0.3–0.6 | 100 | 91.9 | 6.9 ± 0.03 |
| | | 0.5–0.5 | 100 | 100 | 6.6 ± 0.02 |
| | 12% | 0.5–0.5 | 100 | 99 | 4.7 ± 0.02 |

$(L)_{m,encap.PCM}$—Latent heat of melting for encapsulated PCM (J/g)
$(L)_{s,encap.PCM}$—Latent heat of solidification for encapsulated PCM (J/g)
$(L)_{m,PCM}$—Latent heat of melting for PCM (J/g)
$(L)_{s,PCM}$—Latent heat of solidification for PCM (J/g)
n—Encapsulation ratio (%)
ε—Encapsulation efficiency (%)

Several researchers [26,46,47] attempted the encapsulation of coconut oil in microfibers. In the existing literature [26], biomass microfibers with coconut oil encapsulated in the core resulted in melting and solidification temperatures, and enthalpies for the core material of $T_m$ = 22 °C, $T_{c1}$ = 14 °C, $T_{c2}$ = 8 °C, and $\Delta H_m$ = 134.9 J/g, $\Delta H_c$ = 64.7 J/g. The average fiber diameters were 3 ± 1 μm for the cylindrical region of the fiber, and 8 ± 4 μm for the biconical region of the fiber [26]. In another study, coconut oil was successfully encapsulated in PCL gel nanofibers with an efficiency of 60% [46] and 300 to 370 nm mean diameter. Moreover, the melting and crystallization points of the coconut-oil-loaded sample were 25 and 3 °C [46]. Saravana Kumar Jaganathan et al. [47] examined electrospun polyurethane/virgin coconut oil composites, and the fiber's diameter was in the range of 886 ± 207 nm.

A coaxial electrospinning setup was adjusted in the laboratory, core-shell fibers were formed for the oils mixture, and the four commercial PCMs were mixed with the two oils in a PCL shell. The DSC thermographs of tamanu oil (Figure 9a), coconut oil (Figure 9b), and tamanu oil mixed with coconut oil (Figure 9c) in the core of the fiber display two peaks for the PCM and the shell material. The melting or solidification enthalpy (Figure 9a) for tamanu oil was stable (Table 6) for the different flow rates and polymer concentrations. In the case of coconut oil, the electrospun fiber matrix examined in this study (Figure 9b) showed a narrower phase change temperature range (23 to 8 °C) and an increased enthalpy of melting and solidification at 49 to 47 kJ/kg for the case of 12% PCL shell. In the case of tamanu–coconut oil mixture, the electrospun fiber matrix with 9% PCL shell and 0.5/0.5 ml/h core/shell flow rate, enthalpy of 44–35 kJ/kg was observed. In the four cases of commercial PCM materials (RT15 (Figure 9d), RT18 (Figure 9e), Pure Temp 15 (Figure 9f), and Pure Temp 18 (Figure 9g) mixed with the two oils, the melting and solidification curves followed the same trend as that of the curves presented in [40] for electrospun fibers with commercial PCMs alone. Fiber mats with the mixture of RT15 and the two oils in the core (Figure 9d) displayed a phase-change temperature range of 13–9 °C, while the mixture of RT18 with the two oils displayed a 13–2 °C PCM temperature range. In the two mixtures of the organic paraffins with the renewable oils, a higher enthalpy was observed in the polymer concentration of 9% *w/v* with a 0.5/0.5 mL/h core/shell flow rate. The mixture of organic nonparaffinic PCM PT15 with the two oils (Figure 9f) resulted in the melting or solidification temperature range of 8–0 °C, and the highest enthalpies, in this case, 43 kJ/kg for melting and 58 kJ/kg for solidification observed for the polymer concentration of 12% *w/v* and a 0.5/0.5 mL/h core/shell flow rate. Lastly, organic nonparaffinic PCM PT18 demonstrated a melting solidification temperature range from 6 to −6 °C, and exhibited the highest enthalpies of 24 and 34 kJ/kg for a polymer concentration of 9% *w/v* and a 0.5/0.5 mL/h core/shell flow rate. The highest encapsulation ratio n and encapsulation efficiency ε were observed in the cases of polymer concentration of 9% *w/v* and 0.5/0.5 mL/h core/shell flow rate. The equation used to calculate the encapsulation ratio and the efficiency given in [40] indicates that the latent heat of melting for the solution encapsulated in the fiber core for organic paraffins RT15 and RT18 mixed with the two oils is equal to the latent heat of melting the fiber. That being the case, after calculating the encapsulation ratio in organic paraffins RT15 and RT18 with the oil mixture, the outcome was around 100%. This high percentage indicates that the core and fiber materials were melted together in the DSC testing procedure. The average mean diameter of all examined fibers was from 0.1 to 12.8 μm. SEM images show that the 12% *w/v* and 0.5–0.5 mL/h flow rate PT15, tamanu oil, and coconut oil fibers were homogeneous and remained stable, with a mean diameter of 1.5 μm and an encapsulation efficiency of 67.2%.

In future work, a uniform layer of electrospun fibers should be fabricated, applied, and tested in the LHTES system to evaluate if the stored energy is sufficient for construction applications.

## 4. Conclusions

Tamanu oil and coconut oil were studied for potential use as PCM materials in bulk, emulsion, and encapsulated fiber forms. As an outcome of the DSC analysis, bulk coconut oil was classified as a possible new PCM candidate for thermal energy storage applications

with latent heats of 50 and 56 kJ/kg. Tamanu oil and coconut oil emulsions with water and Tween 20 and 80 exhibited lower latent heats than the bulk materials did. The mixture of tamanu oil with coconut oil at a ratio of 70/30 (CO/TO) demonstrated higher latent heats compared to the mixtures with other ratios. The addition of 50% of RT18, a commercially available PCM, to 70/30 mixture of coconut and tamanu oil yielded electrospun fibers with the best latent heat of melting and solidification of 63.8 and 57.6 kJ/kg, respectively. Overall, the developed procedure of coaxial electrospinning with a PCL shell was demonstrated to be suitable for the efficient encapsulation of the PCM in the fibers. In the cases of RT15–tamanu oil–coconut oil and RT18–tamanu oil–coconut oil, the latent heat of the PCM fiber was equal to the latent heat of the encapsulated PCM emulsion. The successful production of electrospun PCM fiber mats with encapsulated oils of biological origin is an important step towards energy-saving and environmentally friendly construction materials.

**Author Contributions:** Conceptualization, E.P. and P.F.; methodology, E.P. and A.A.; software, E.P.; validation, E.P., L.G. and P.F.; formal analysis, E.P.; investigation, E.P., P.F. and A.A.; resources, P.F.; data curation, P.F. and E.P.; writing—original draft preparation, E.P.; writing—review and editing, E.P., L.G., P.F. and A.A.; supervision, E.P., P.F. and A.A.; project administration, E.P.; funding acqui-sition, A.A. All authors have read and agreed to the published version of the manuscript.

**Funding:** The authors acknowledge the support provided by ELFORSK, a research and development program administrated by Danish Energy.

**Institutional Review Board Statement:** Not applicable.

**Informed Consent Statement:** Not applicable.

**Data Availability Statement:** Data are available upon request.

**Conflicts of Interest:** The authors declare no conflict of interest.

### Nomenclature

| Parameter | Description | Unit |
|---|---|---|
| PCM | Phase change material | - |
| TES | Thermal energy storage | - |
| PCL | Polycaprolactone | - |
| SDS | Sodium dodecyl sulfate | - |
| DSC | Differential scanning calorimetry | - |
| $(L)_{m,encap.PCM}$ | Latent heat of melting for encapsulated PCM | (J/g) |
| $(L)_{s,encap.PCM}$ | Latent heat of solidification for encapsulated PCM | (J/g) |
| $(L)_{m,PCM}$ | Latent heat of melting for PCM | (J/g) |
| $(L)_{s,PCM}$ | Latent heat of solidification for PCM | (J/g) |
| $T_m$ | Melting temperature | (°C) |
| $T_s$ | Solidification temperature | (°C) |
| n | Encapsulation ratio | (%) |
| $\varepsilon$ | Encapsulation efficiency | (%) |
| $w/v$ | Weight/volume | (% g/mL) |
| $v/v$ | Volume/volume | (% mL/mL) |

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
