# Peer review of "Thermal Properties of Novel Phase-Change Materials Based on Tamanu and Coconut Oil Encapsulated in Electrospun Fiber Matrices"

_sustainability, doi:10.3390/su14127432_

Round 1

Reviewer 1 Report

Good work  towards energy-saving and environmentally friendly. This type of more and more research is required to do for future generation health and well being.

Author Response

Point 1: Good work  towards energy-saving and environmentally friendly. This type of more and more research is required to do for future generation health and well being.

Response 1: Thank you for your comments. We appreciate the time and effort that you have dedicated to provide your valuable feedback on our manuscript.

Reviewer 2 Report

Major remarks:

1.      What is the problem and how can nanofibers containing PCM help to solve this problem?

2.      Basically, why are nanofibers produced?

3.      Given the small volume of materials, is the amount of energy stored sufficient for construction applications?

Minor remarks:

1.      The variables, hypothesis and solutions should be mentioned in last paragraph of introduction.

2.      The introduction/ method and materials/ results and discussion are not prepared in scientific manner. For example:

·         2/21 line 57: the paragraph "For the current study, we have selected coconut oil from coconut palm tree …. " should be transferred to method and materials section.

·         10/21 line 273 "Four commercial PCM (RT15, RT18, PT15, and PT18) were mixed …" should be transferred to method and materials.

Author Response

Response to Reviewer 2

Major remarks:

  1. What is the problem and how can nanofibers containing PCM help to solve this problem?

Response 1: The current work aims to identify the thermal properties of electrospun nanofibers as an alternative to PCM in liquid-solid form. In future work, a layer of nanofibers containing PCM could replace traditional PCM in LHTES applications. In the last paragraph of the introduction, a problem description and explanation of the potential use of PCM nanofibers are added.

  1. Basically, why are nanofibers produced?

Response 2: As mentioned before, it is aimed to produce a thick layer of nanofibers as an alternative to PCM in pure form for future use in LHTES systems. In the last paragraph of the introduction, a problem description and explanation of the potential use of PCM nanofibers are added.

  1. Given the small volume of materials, is the amount of energy stored sufficient for construction applications?

Response 3: A layer of electrospun fiber matrices should be fabricated to evaluate if the energy stored is sufficient for construction applications. The PCM fibers layer could be encapsulated in a LHTES system in future work. In the last paragraph of results and discussion a statement is added.

Minor remarks:

  1. The variables, hypothesis and solutions should be mentioned in last paragraph of introduction.

Response 1: The last paragraph of introduction was modified and contain the variables and solutions analyzed in the manuscript.

  1. The introduction/ method and materials/ results and discussion are not prepared in scientific manner. For example:
  • 2/21 line 57: the paragraph "For the current study, we have selected coconut oil from coconut palm tree …. " should be transferred to method and materials section.
  • 10/21 line 273 "Four commercial PCM (RT15, RT18, PT15, and PT18) were mixed …" should be transferred to method and materials.

Response 2:

  1. The paragraph "For the current study, we have selected coconut oil from coconut palm tree …. " was transferred to methods and materials sections. Both introduction and materials and methods sections were modified.
  2. The sentence "Four commercial PCM (RT15, RT18, PT15, and PT18) were mixed …" was transferred to methods and materials.

Reviewer 3 Report

Row 35:

It would be helpful to specify the basic material requirements for TES. Or are only the heat of fusion and phase change temperature essential properties? What is expected of the materials concerning the applied methods?

 Row 43:

Is it really correct, compared to CO2 emissions from agricultural production (primarily animal breeding? Try to find another cited source to confirm this statement.

 Chapter 2.1:

What crucibles were used? I assume aluminium, but open, hermetic, pinhole, etc. Include also atmosphere during measurement (air, nitrogen).

 Figure 2:

I'm a bit confused when assigning individual curves for each sample. You mentioned different ratios (1:1, 1:5) and different surfactants. so what exactly does the labeling 1 (1 1) 2 t20; 1 (1 1) 5 t20; 1 (2 1) 3 t20 mean? Specific labeling of the samples with the exact composition (ratios and additives) in a uniform table would help to better orient the results. Thus, a detailed search is required to find what specific samples are involved and cannot be devoted to concentrating on the results.

 Figure 3:

I'm a bit confused when assigning individual curves for each sample. You mentioned different ratios (1:1, 1:5) and different surfactants. so what exactly does the labeling 1 (1 1) 2 t20; 1 (1 1) 5 t20; 1 (2 1) 3 t20 mean? Specific labeling of the samples with the exact composition (ratios and additives) in a uniform table would help to better orient the results. Thus, a complicated search is required to find what specific samples are involved and cannot be devoted to concentrating on the results. A surprising explanation is provided in the caption of Fig. 4: Water:(Oil:Emulsifier); a) 1:(1:1)2, b) 1:(1:1)5, c) 1:(2:1)3. A surprising explanation is provided in the caption of Fig. 4: Water:(Oil:Emulsifier); a) 1:(1:1)2, b) 1:(1:1)5, c) 1:(2:1)3.

 Consider the location of the sample specification before presenting the results.

 Equations 1 and 2:

Missing description of individual variables in the equation used.

 The form of Fig. 10, 11, and 12  is not well chosen and forces the reader to continually flip through the images to assign a particular sample modification.

Please complete the opinion on fire safety and materials used?

Does the presence of moisture in the materials play any role? What about applying the preheating to reduce water content, humidity in oil, etc.?

Author Response

Response to Reviewer 3

  1. Row 35: It would be helpful to specify the basic material requirements for TES. Or are only the heat of fusion and phase change temperature essential properties? What is expected of the materials concerning the applied methods?

Response 1: Some basic thermal properties of PCM suitable for TES applications are density, thermal conductivity, phase change temperature, the heat of fusion, and the supercooling degree. In the current research, we have identified the ability of PCM to store and release large amounts of energy. Thus we have focused on the latent heat of fusion and the phase change temperature of the PCM. The experimental values of the phase change temperature and heat of fusion of all materials examined in their pure form were compared to data provided by the manufacturers, and the relative error was calculated in [1].

  1. Row 43: Is it really correct, compared to CO2 emissions from agricultural production (primarily animal breeding? Try to find another cited source to confirm this statement.

Response 2: Thank you for your comment. In the existing literature, it is not proven that crude plant and animal oil are associated with lower CO2 emissions. For this reason, it was decided to delete the statement.

  1. Chapter 2.1: What crucibles were used? I assume aluminium, but open, hermetic, pinhole, etc. Include also atmosphere during measurement (air, nitrogen).

Response 3: The DSC tests were performed using a DSC Q2000 instrument with T-zero thermocouples. This DSC equipment offers 50 positions for samples and a temperature range of -180 to 725. A conventional empty aluminum crucible was used as a blank for the reference sample. The DSC equipment includes a furnace where a reference sample and the examined sample are heated or cooled under controlled temperature variations. Nitrogen was used as inert purge gas and air with a controlled flow rate of 50ml/min.

  1. Figure 2: I'm a bit confused when assigning individual curves for each sample. You mentioned different ratios (1:1, 1:5) and different surfactants. so what exactly does the labeling 1 (1 1) 2 t20; 1 (1 1) 5 t20; 1 (2 1) 3 t20 mean? Specific labeling of the samples with the exact composition (ratios and additives) in a uniform table would help to better orient the results. Thus, a detailed search is required to find what specific samples are involved and cannot be devoted to concentrating on the results.

Response 4: Thank you for your suggestion. We have added detailed labeling of the samples in Table 1.

  1. Figure 3: I'm a bit confused when assigning individual curves for each sample. You mentioned different ratios (1:1, 1:5) and different surfactants. so what exactly does the labeling 1 (1 1) 2 t20; 1 (1 1) 5 t20; 1 (2 1) 3 t20 mean? Specific labeling of the samples with the exact composition (ratios and additives) in a uniform table would help to better orient the results. Thus, a complicated search is required to find what specific samples are involved and cannot be devoted to concentrating on the results. A surprising explanation is provided in the caption of Fig. 4: Water:(Oil:Emulsifier); a) 1:(1:1)2, b) 1:(1:1)5, c) 1:(2:1)3. A surprising explanation is provided in the caption of Fig. 4: Water:(Oil:Emulsifier); a) 1:(1:1)2, b) 1:(1:1)5, c) 1:(2:1)3.Consider the location of the sample specification before presenting the results.

Response 5: Thank you for your suggestion. We have added detailed labeling of the samples in Table 1.

  1. Equations 1 and 2: Missing description of individual variables in the equation used.

Response 6: A description of the individual variables was added after equations 1 and 2.

  1. The form of Fig. 10, 11, and 12 is not well chosen and forces the reader to continually flip through the images to assign a particular sample modification.

Response 7:  Your suggestion was taken into account, and we have incorporated an indication of the exact material below every figure.

  1. Please complete the opinion on fire safety and materials used?

Response 8: Ingenious strategies, e.g., additives and flame retardant elements, shall be applied to eliminate the moderate-high flammability of organic paraffins and fatty acids. Further research shall be conducted to analyze the flammability of PCM.

  1. Does the presence of moisture in the materials play any role? What about applying the preheating to reduce water content, humidity in oil, etc.?

Response 9: To control both indoor temperature and humidity, phase change humidity control materials (PCHCM) have been analyzed by Chen et al. [2]. In the present experimental research, the presence of moisture in the case of pure tamanu oil may have affected the material’s thermal properties. In future research, strategies for reducing humidity in oil shall be applied.

[1]      E. Paroutoglou, A. Afshari, P. Fojan, and G. Hultmark, “Investigation of Thermal Behavior of Paraffins, Fatty Acids, Salt Hydrates, and Renewable Based Oils as PCM,” Proc. 14th Int. Renew. Energy Storage Conf. 2020 (IRES 2020), vol. 6, 2021, doi: 10.2991/ahe.k.210202.006.

[2]      Z. Chen and M. Qin, “Preparation and hygrothermal properties of composite phase change humidity control materials,” Appl. Therm. Eng., vol. 98, pp. 1150–1157, 2016, doi: 10.1016/j.applthermaleng.2015.12.096.

Reviewer 4 Report

Comments on “Thermal properties of novel phase-change materials based on  tamanu and coconut oil encapsulated in electrospun fiber matrices”

Dear Authors,

The paper must be significantly improved. Please consider the following remarks:

Major comments:

(1) Abstract does not inlude any specific results. What is the main scientific issue which you concern in your study.

(2) Keywords. Please expand list / please improve - Authors should make better use of this section to allow the article to be found on search engines.

(3) Line 47-55: please add some main results/conclusions from referenced papers. I do not understand the reason why you mentioned “they achieved an energy reduction of 4% in cooling for the building”.  You should be more precisely, for example, you can compare using kWh per kg per year

(4) Line 105-107 Please add scientific novelty

(5) Line 136 – 137, 171-172, 225-226: “The expanded uncertainty for each experimental measurement was ± 0.1%.” How did you obtain above-mentioned uncertainty?

(6) Line 350-358. Please add more comparable datas.

(7) Line 409-410: “shell was demonstrated suitable for efficient encapsulation of the PCM in the fibers.” What does “efficient mean? Please explain.

Minor comments (answers are not necessary):

(1) Line 70-71: “application with coconut oil as PCM in a container and reduced 0.82-1.29ºC the air temperature.” It is not comparable.

(2) Line 118 From the SI Brochure, §5.3. 3: "The numerical value always precedes the unit, and a space is always used to separate the unit from the number." The same approach I suggest applying also in the rest of the manuscript.

(3) Line 132: “emulsions emulsions” ?

(4) Line 182, 185: Please improve the way of calling references

(5) Line 402 “degradation s during” ?

(6) Line 415. Please improve the Nomenclature table. I suggest following name of column: parameter/name; description; unit

(7) Please improve Reference part in line with MDPI template

Author Response

Response to Reviewer 4

Major comments:

  1. Abstract does not inlude any specific results. What is the main scientific issue which you concern in your study.

Response 1: Thank you for your comment. A summary of the most significant outcome of the study was added to the abstract.

  1. Please expand list / please improve - Authors should make better use of this section to allow the article to be found on search engines.

Response 2: We agree with your suggestion and have tried to improve the list of keywords.

  1. Line 47-55: please add some main results/conclusions from referenced papers. I do not understand the reason why you mentioned “they achieved an energy reduction of 4% in cooling for the building”. You should be more precisely, for example, you can compare using kWh per kg per year

Response 3: As you have suggested, the main results of referenced papers were added. The sentence “they achieved an energy reduction of 4% in cooling for the building” was rephrased.

  1. Line 105-107 Please add scientific novelty

Response 4: Thank you for your suggestion. We believe that the scientific novelty can be addressed in the novel fiber matrices of oil mixtures with commercial PCM paraffins produced by the co-axial electrospinning technique.

  1. Line 136 – 137, 171-172, 225-226: “The expanded uncertainty for each experimental measurement was ± 0.1%.” How did you obtain above-mentioned uncertainty?

Response 5: The measurements are not repeated, and the calorimetric accuracy is 0.05%; thus, for 95.4% of accuracy (k=2), the final expanded uncertainty is 0.1% for each experimental measurement.

  1. Line 350-358. Please add more comparable datas.

Response 6: In this section, we decided to present relevant studies examining electrospun fiber with coconut oil and tamanu oil encapsulated in the fiber's core. Only a few studies in the literature address coconut oil encapsulated in the core of electrospun fibers. Unfortunately, we have not found relevant studies where tamanu oil is encapsulated in fiber’s core.

  1. Line 409-410: “shell was demonstrated suitable for efficient encapsulation of the PCM in the fibers.” What does “efficient mean? Please explain.

Response 7: In the process of fabrication of PCM electrospun fiber matrices, several polymers have been tested to construct the fiber’s shell. PCL in DCM was selected as a suitable material for the encapsulation of PCM since the electrospun fiber kept its structure and did not melt after fabrication.

Minor comments (answers are not necessary):

  1. Line 70-71: “application with coconut oil as PCM in a container and reduced 0.82-1.29ºC the air temperature.” It is not comparable.

Response 1: Thank you for your suggestion. We have considered your comment and modified the content in lines 69-71.

  1. Line 118 From the SI Brochure, §5.3. 3: "The numerical value always precedes the unit, and a space is always used to separate the unit from the number." The same approach I suggest applying also in the rest of the manuscript.

Response 2: Thank you for your suggestion. The proposed approach has been used in the manuscript.

  1. Line 132: “emulsions emulsions” ?

Response 3: Spelling mistake corrected in the submitted manuscript.

  1. Line 182, 185: Please improve the way of calling references

Response 4: References have been adjusted according to MDPI template.

  1. Line 402 “degradation s during” ?

Response 5: Spelling mistake corrected in the submitted manuscript.

  1. Line 415. Please improve the Nomenclature table. I suggest following name of column: parameter/name; description; unit

Response 6: The Nomenclature table was improved according to your suggestion in the submitted manuscript.

  1. Please improve Reference part in line with MDPI template

Response 7: References have been adjusted according to MDPI template.

Round 2

Reviewer 2 Report

The manuscript can be published in this condition.

Author Response

Response 2: Thank you for your comments. We appreciate the time and effort that you have dedicated to provide your valuable feedback on our manuscript.

Reviewer 4 Report

Please improve nomenclature and abbreviations table:

1) Please use alphabetical order

2) Please add all nomenclature: H, CO/TO, LHTES etc

Author Response

Response 1: Thank you for your comment. We will improve nomenclature table according to your suggestion.